# *JUN* mRNA translation regulation is mediated by multiple 5′ UTR and start codon features

**Angélica M. González-Sánchez[1], Eimy A. Castellanos-Silva[2], Gabriela Díaz-Figueroa[3], Jamie H. D. Cate[3]***

1 Comparative Biochemistry Graduate Program, University of California, Berkeley, Berkeley, CA, United States of America, 2 Department of Biochemistry and Molecular Biology, University of California, Davis, Davis, CA, United States of America, 3 Department of Molecular and Cell Biology, University of California, Berkeley, Berkeley, CA, United States of America

* j-h-doudna-cate@berkeley.edu

## Abstract

Regulation of mRNA translation by eukaryotic initiation factors (eIFs) is crucial for cell survival. In humans, eIF3 stimulates translation of the *JUN* mRNA which encodes the transcription factor JUN, an oncogenic transcription factor involved in cell cycle progression, apoptosis, and cell proliferation. Previous studies revealed that eIF3 activates translation of the *JUN* mRNA by interacting with a stem loop in the 5′ untranslated region (5′ UTR) and with the 5′-7-methylguanosine cap structure. In addition to its interaction site with eIF3, the *JUN* 5′ UTR is nearly one kilobase in length, and has a high degree of secondary structure, high GC content, and an upstream start codon (uAUG). This motivated us to explore the complexity of *JUN* mRNA translation regulation in human cells. Here we find that JUN translation is regulated in a sequence and structure-dependent manner in regions adjacent to the eIF3-interacting site in the *JUN* 5′ UTR. Furthermore, we identify contributions of an additional initiation factor, eIF4A, in *JUN* regulation. We show that enhancing the interaction of eIF4A with *JUN* by using the compound Rocaglamide A (RocA) represses *JUN* translation. We also find that both the upstream AUG (uAUG) and the main AUG (mAUG) contribute to *JUN* translation and that they are conserved throughout vertebrates. Our results reveal additional layers of regulation for *JUN* translation and show the potential of *JUN* as a model transcript for understanding multiple interacting modes of translation regulation.

## Introduction

Protein translation is one of the most energetically expensive cellular processes and is highly regulated, especially during translation initiation [1–5]. Translation initiation is a complex process which regulates expression of eukaryotic genes and employs over a dozen eukaryotic translation initiation factors (eIFs) [6–9]. These include eIF1, eIF1A, eIF3, eIF5, eIF2 and the eIF4F complex, which is composed of eIF4E, eIF4A and eIF4G [7, 9]. During eukaryotic translation initiation, a ternary complex made up of initiator methionyl-tRNA (Met-tRNA$_i$), eIF2, and GTP is formed [10, 11]. The 43S pre-initiation complex (PIC) then comes together by

**Data Availability Statement:** All data are in the manuscript and/or supporting information files.

**Funding:** This work was funded by grants from the National Institute of General Medical Sciences

(R01-GM065050 and R35-GM148352) to J.H.D.C. https://reporter.nih.gov/ The funders played no role in the study design, data collection and analysis, decision to publish, or preparation of the manuscript.

**Competing interests:** The authors have declared that no competing interests exist.

recruitment of the ternary complex, the 40S ribosomal subunit, and eukaryotic initiation factors 1, 1A, 3 and 5 [12–16]. After adopting an open conformation, the 43S PIC joins eIF4F at the mRNA 5′ cap in order to recruit the mRNA to form the 48S PIC [11]. This newly formed 48S PIC is then capable of scanning the mRNA through its 5′ untranslated region (5' UTR) until it locates a start codon [17]. Once the start codon is recognized, several initiation factors are released in order for the ribosome to begin elongation [9, 11]. During initiation, the roles of several eIFs have been linked to translation regulation of subsets of mRNAs. For example, experiments performed in human cells revealed that eIF3 regulates the translation of specific mRNAs by direct interactions [18–20]. These eIF3-mRNA interactions are important for homeostasis but also play essential roles upon nutrient deprivation and drive the integrated stress response, among other functions [18–30]. eIF4A, an RNA helicase, has also been associated with translation regulation of a subset of mRNAs in human cells, more specifically by unwinding 5′ UTRs that are highly structured and polypurine rich and many of which are related to cell-cycle progression and apoptosis [31–34]. Moreover, eIF1 and eIF5 play important roles in the selection of translational start sites, depending not only on the AUG translational context, but also on the abundance of these initiation factors and specific cellular conditions [35–41].

Translation initiation factor eIF3 is a crucial player in protein expression regulation through its roles in bridging the 43S PIC and eIF4F complexes, and also by performing specialized regulatory roles [15, 42, 43]. eIF3 specifically binds to and regulates translation of a subset of mRNAs, many of which are involved in cell cycle regulation, cell growth, differentiation, and other crucial cellular functions. The interaction between eIF3 and mRNAs was shown to be mediated by RNA structural elements in the 5′ UTR of specific mRNAs in human embryonic kidney (HEK293T) cells and to cause translational activation or repression of these mRNAs [18]. eIF3 has also been shown to have cell-specific regulatory roles in T cells, with eIF3 interactions throughout the entire length of the transcript for specific mRNAs, such as the ones encoding the T cell receptor alpha and beta subunits (*TCRA* and *TCRB*, respectively), mediating a translational burst essential for T cell activation [20]. In yeast, eIF3 has also been linked to mRNA recruitment and scanning as a mediator of mRNA-PIC interactions [44–46]. Furthermore, in zebrafish eIF3 subunit H (EIF3H) was shown to regulate translation of mRNAs encoding the eye lens protein crystallin during embryogenesis [47]. These examples demonstrate that eIF3 plays a variety of mRNA-specific regulatory roles.

One of the reported eIF3-target mRNAs in human cells, *JUN*, encodes the transcription factor JUN, also known as c-Jun, which regulates gene expression in response to different stimuli [48, 49]. As a component of the activator protein-1 (AP-1) complex, JUN regulates transcription of a large number of genes and acts mainly as a transcriptional activator [50]. JUN is therefore highly involved in various cellular processes including cell proliferation, apoptosis, tumorigenesis, and it was the first oncogenic transcription factor discovered [49, 51, 52]. Regulation of JUN expression is particularly important because its downregulation can lead to cell cycle defects and its upregulation can lead to accelerated cell proliferation, which occurs in some cancers [53]. Therefore, it is not surprising for JUN expression regulation to be complex and to occur at both the transcriptional and translational levels. At the transcriptional level, *JUN* mRNA expression is regulated by its own protein product, which binds a high-affinity AP-1 binding site in the *JUN* promoter region and in turn induces its transcription [54–56]. *JUN* expression regulation at the translational level is mediated by its mRNA interaction with eIF3. Binding of eIF3 subunits EIF3A, EIF3B, EIF3D, and EIF3G to a stem loop in the *JUN* 5′ UTR results in activation of translation [18]. Moreover, eIF3 subunit D (EIF3D) acts as a 5′ cap-binding protein on the *JUN* mRNA, mediated by a cis-acting RNA element located in the 153 nucleotides immediately downstream of the *JUN* 5′-7-methylguanosine cap structure [19].

This RNA element is also thought to block recruitment of the eIF4F complex [19]. *JUN* expression regulation at the translational level has also been shown to be affected by m⁶A methylation by METTL3 in its 3′ UTR and by contributions of an RNA structural element which activates its translation in glioblastoma [53, 57].

*JUN* possesses a longer than average 977-nucleotide 5′ UTR that is highly GC rich. Due to its length and complexity, *JUN*'s 5′ UTR might present additional layers of translational regulation of its mRNA through novel structural and/or sequence elements. Previously reported involvement of several initiation factors, including eIF3 and eIF4A, in the recruitment of mRNAs with long and structurally complex 5′ UTRs further supports a 5′ UTR-mediated mechanism for *JUN* translation regulation and suggests that additional factors may be involved in JUN regulation [58]. For example, *JUN* was recently shown to be sensitive to RocA, an anti-cancer drug that clamps eIF4A onto specific polypurine sequences in the 5′ UTRs of a subset of mRNAs [33, 34]. However, the implications of this interaction on *JUN* translation have not been previously evaluated. JUN also possesses two potential translational start sites, an upstream start codon (uAUG) located 4 codons upstream of the main start codon (mAUG). However, translational start site selection for the *JUN* mRNA has not been previously explored. Therefore, we further investigated *JUN* translation regulation in human cells by exploring different regions of the *JUN* 5′ UTR and how mRNA features and the interaction of initiation factors in these regions contribute to *JUN* translation. Firstly, we applied mutagenesis to the *JUN* 5′ UTR near the eIF3 binding site. We also further investigated the contributions of eIF4A to *JUN* translation both by mRNA mutagenesis and through cellular treatment with RocA. Finally, we explored how the translational context of both of the *JUN* start codons affect start site selection. Our results demonstrate that *JUN* translation regulation is a complex multi-layered process that involves various initiation factors, including eIF3 and eIF4A, and mRNA features such as secondary structures in the 5' UTR.

## Results

### *JUN* translation is regulated by 5' UTR sequence and structural elements

Binding of eIF3 to a stem-loop in the 5' UTR of the *JUN* mRNA leads to its translational activation [18]. Mutations in this stem loop have been shown to disrupt the interaction with eIF3 and to repress *JUN* translation [18]. However, the effects of other mutations in the *JUN* 5′ UTR remain to be explored. We first tested whether mutations in other regions within and near the *JUN*-eIF3 interacting stem loop (SL) affect *JUN* translation. We generated mRNA reporter constructs containing the full-length *JUN* 5′ UTR and Nanoluciferase (Nluc) coding sequence (CDS) that included mutations in a 208 nucleotide (nt) SL proximal region whose secondary structure was previously determined [18] by selective 2'-hydroxyl acylation analyzed by primer extension, also known as SHAPE (Fig 1A, SHAPE). All of the mutations disrupt either the secondary structure or the sequence of highly structured regions within the SL proximal region (Fig 1B). For each of these constructs transfected into HEK293T cells, together with an mRNA reporter with the Hemoglobin Beta Subunit (*HBB)* 5′ UTR and a Firefly luciferase (Fluc) CDS as an internal control, we assessed translation using luciferase assays.

As expected, deletion of the *JUN*-eIF3 interacting stem loop (Fig 1C, mutant ΔSL) significantly represses *JUN* reporter translation when compared to the WT construct. Mutations to SL loop nucleotides C128-U129, previously shown to be unreactive by SHAPE mapping *in vitro* and therefore likely to be involved in RNA-RNA contacts, also significantly affected *JUN* reporter translation, with U129G dramatically increasing translation (Fig 1C, mutant A) [18]. Interestingly, replacing the SL loop with a much smaller and possibly more stable UUCG tetra-loop substantially increased *JUN* reporter translation (Fig 1C, mutant E) [59]. However,

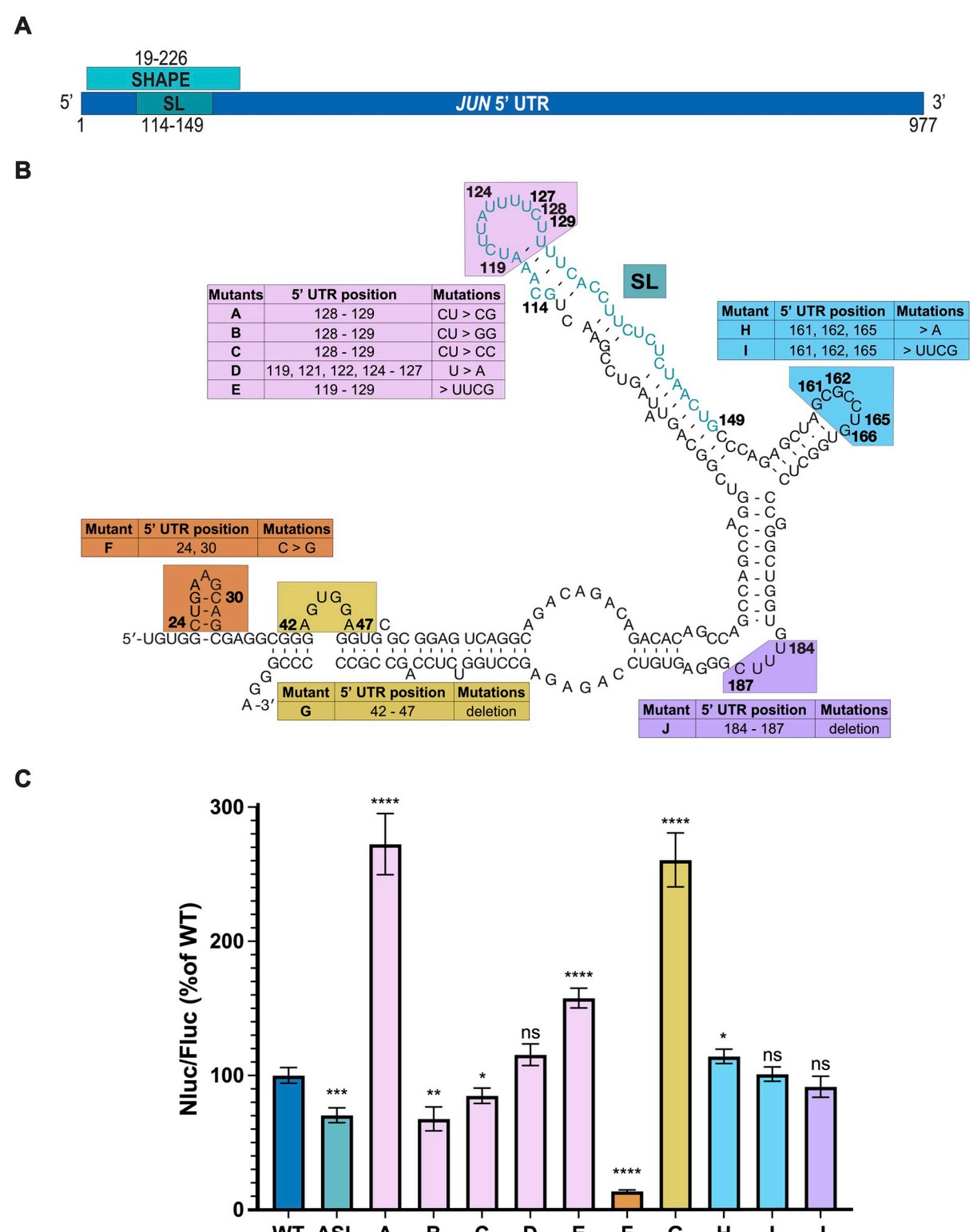

**Fig 1. *JUN* translation is regulated by 5' UTR sequence and structural elements.** (A) Depiction of the full *JUN* 5' UTR. The locations of the 208-nt region studied by SHAPE (SHAPE) and the eIF3-interacting stem loop (SL) are marked, along with the nucleotides involved in each region. (B) Secondary structure of the 208-nt region in the *JUN* 5' UTR mapped by SHAPE is shown. Nucleotides are numbered according to their position in the 5' UTR. Mutant *JUN* 5' UTR mRNA constructs and their corresponding mutations are described in their associated tables. (C) Luminescence measured from HEK293T cells transfected with the *JUN* 5' UTR reporter mRNAs expressing Nanoluciferase (Nluc).

Translation was assessed using a dual-luciferase assay and normalized to a control mRNA harboring an *HBB* 5′ UTR and a Firefly luciferase (Fluc) CDS. Nluc/Fluc ratios were normalized to the WT *JUN* 5′ UTR, set as 100%. Technical triplicates for each biological replicate, and a total of at least three biological replicates were taken for each measurement. P values determined using a one-sample t test versus a hypothetical value of 100 are shown as follows: *p ≤ 0.05, **p ≤ 0.01, ***p ≤ 0.001, ****p ≤ 0.0001. The mean value of the replicates and standard error of the mean are shown.

replacing all of the U's with A's in the loop sequence had little effect on *JUN* translation (Fig 1C, mutant D). As a whole, these findings support the importance of the SL loop in *JUN* translation regulation, yet reveal a complexity in its role maintaining and stabilizing the secondary structure of the SL region. Mutations in other structured regions of the *JUN* 5′ UTR near the eIF3 binding site also significantly affected *JUN* translation. For example, disrupting the stem loop between nucleotides 23 and 33 with point mutations in nucleotides 24 and 30 repressed *JUN* reporter translation (Fig 1C, mutant F). By contrast, deleting the bulge loop formed by nucleotides 42 to 47 increased *JUN* reporter translation (Fig 1C, mutant G). These findings suggest that these secondary structure features in the *JUN* 5′ UTR outside the originally identified eIF3 binding site play opposing roles in regulating *JUN* translation. However, mutations to two other loop and bulge regions near the SL (nts 160–166 and 184–187) had little or no effect on *JUN* reporter translation (Fig 1C, mutants H-J).

## *JUN* is highly sensitive to RocA treatment

Rocaglamide A (RocA) is an anti-cancer compound that specifically clamps eIF4A onto polypurine sequences in a subset of mRNAs, in an ATP-independent manner. This clamping of eIF4A blocks 43S scanning, leading to premature, upstream translation initiation and reducing protein expression from transcripts containing RocA–eIF4A target sequences [33, 34]. Interestingly, *JUN* is one of the mRNAs identified as highly sensitive to RocA treatment [33]. However, little is known about how promoting or disrupting the *JUN* interaction with eIF4A affects *JUN* translation. To this end, we first transfected *JUN* 5′ UTR and Nluc mRNA reporter constructs designed above (Fig 1B) together with the *HBB* 5′ UTR and Fluc CDS control mRNA, into HEK293T cells and treated these with increasing concentrations of RocA or DMSO (as a negative control). In all the cases we tested, including the WT, ΔSL, and the UUCG tetraloop mutation in the SL loop, treatment with RocA strongly suppressed *JUN* reporter translation (Fig 2A). This effect was not observed for the control Nluc reporter mRNA harboring the *HBB* 5′ UTR, which has not been reported as RocA sensitive. The fact that constructs with mutations that affect the eIF3-interacting stem loop in the *JUN* 5′ UTR were still highly sensitive to RocA treatment suggests that the RocA-mediated effects on the *JUN* 5′ UTR are independent of eIF3 regulation. The persistent repressive trend of RocA treatment on *JUN* translation also suggests that eIF4A serves an important role in *JUN* translation regulation.

RocA-sensitive mRNAs are enriched in the polypurine sequence GAA(G/A) [33]. As shown in Fig 2B, *JUN* possesses 11 of these polypurine sequences across the entire length of its 5′ UTR, with none present in the eIF3-interacting stem loop. In order to evaluate the effect of disrupting these sequences in the *JUN* 5′ UTR, we mutated these polypurine (GAA(G/A)) sequences to the mixed purine/pyrimidine sequence CAAC, previously reported to disrupt RocA-mediated eIF4A binding to mRNAs [34]. Interestingly, the *JUN* reporter mRNAs with these mutations (mutants CAAC or CAAC + ΔSL) remained highly sensitive to RocA (Fig 2C). This indicates that there are additional eIF4A target sequences in the *JUN* 5′ UTR that are not necessarily equivalent to the reported predominant GAA(G/A) motif. Moreover, deleting the eIF3 interacting stem loop, together with the GAA(G/A) mutations (mutant CAAC +

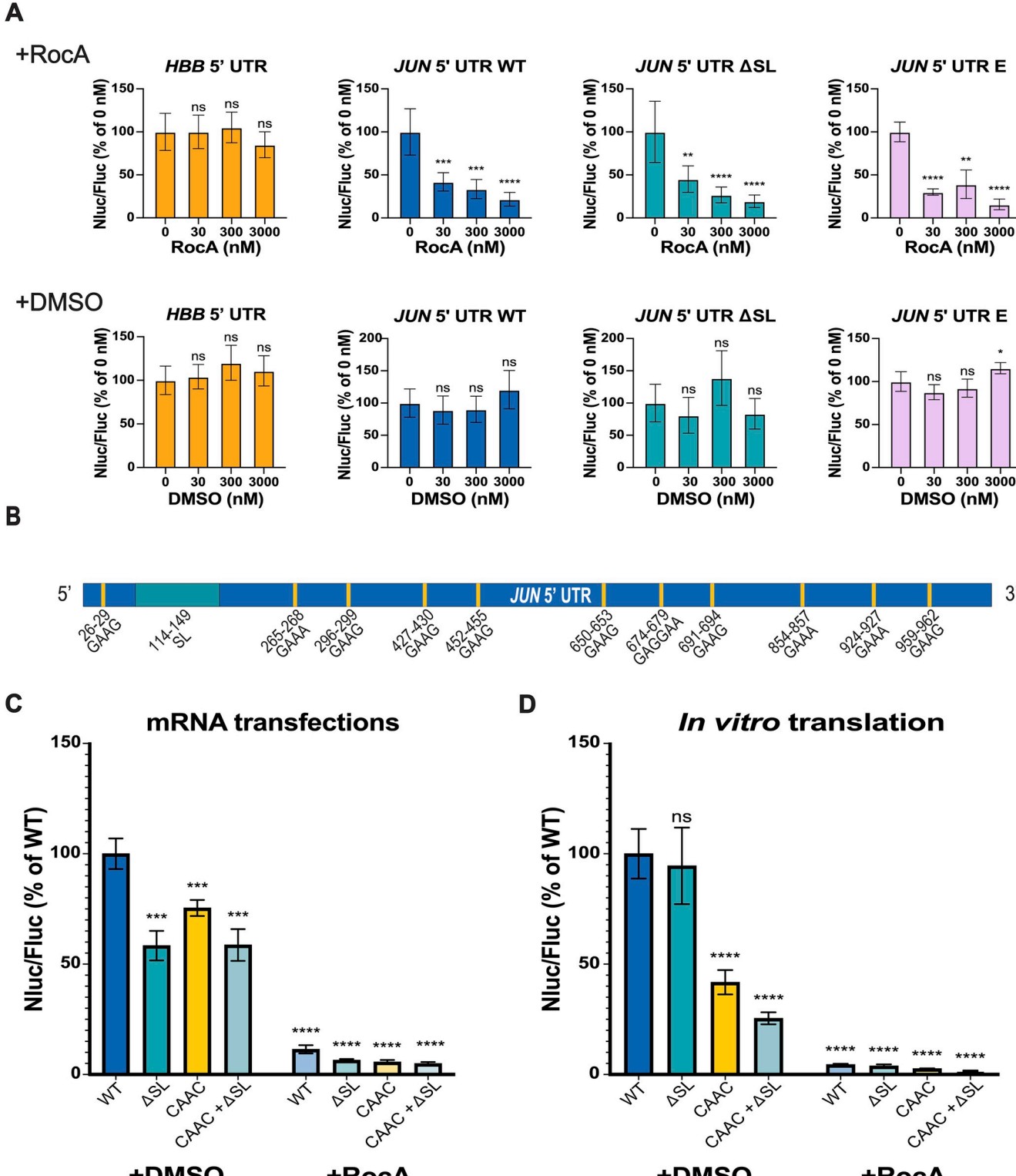

**Fig 2. *JUN* is highly sensitive to RocA treatment.** (A) HEK293T cells co-transfected with *JUN* 5′ UTR and Nluc CDS reporter mRNAs (WT, ΔSL or mutant G, Fig 1) and with an *HBB* 5′ UTR and Fluc mRNA as an internal control, were treated with increasing concentrations of RocA (+RocA) or DMSO control (+DMSO) 3 hours post-transfection, as previously reported [33]. An mRNA with the *HBB* 5′ UTR and Nluc CDS mRNA was also used as a RocA-insensitive control. Translation was assessed using a dual-luciferase assay as in Fig 1. Nluc/Fluc measurements were normalized to the corresponding untreated condition (0 nM RocA) and reported as a percentage of this measurement. (B) The location of polypurine (GAA(G/A)) sequences in the *JUN* 5′ UTR are

indicated with yellow lines. Each of these 11 sequences was mutated to CAAC. (C) Luminescence of HEK293T cells transfected with *JUN* 5′ UTR and Nluc CDS reporter mRNAs (WT, ΔSL, CAAC or CAAC + ΔSL), together with the *HBB* 5′ UTR and Fluc CDS mRNA control. Transfected cells were treated with 300 nM RocA (+RocA) or DMSO (+DMSO) 3 hours post-transfection. Translation was assessed using a dual-luciferase assay as in Fig 1, and Nluc/Fluc measurements were normalized to the WT *JUN* 5′ UTR and Nluc CDS +DMSO measurements, reported as percentages. (D) Luminescence from *in vitro* translation reactions using the *JUN* 5′ UTR and Nluc CDS reporter mRNAs (WT, ΔSL, CAAC or CAAC + ΔSL). Reactions were treated with 300 nM RocA (+RocA) or DMSO (+DMSO). Luminescence values of each mutant were normalized to the WT *JUN* 5′ UTR and Nluc CDS +DMSO measurements and reported as percentages. In panels A, C, and D, technical triplicates for each biological replicate, and a total of at least three biological replicates were taken for each measurement. P values determined using a one-sample t test versus a hypothetical value of 100 are shown as follows: *$p \leq 0.05$, **$p \leq 0.01$, ***$p \leq 0.001$, ****$p \leq 0.0001$. The mean value of the replicates and standard error of the mean are shown.

ΔSL), has no further effect on translation. We observed similar effects with the *JUN* mRNA reporters *in vitro* using HEK293T cell extracts (Fig 2D). Taken together, these results support a model in which eIF4A regulates *JUN* translation in an eIF3 independent manner, pointing to further layers of regulation for *JUN* translation, mediated by additional initiation factors.

## Two start codons contribute to *JUN* translation in cells

Start codon selection regulates the translation of many transcripts [38–41, 60, 61]. Recently, it was reported that the translational context of start codons on transcripts with an upstream open reading frame (uORF) and a main open reading frame (mORF) affects which of these is preferentially selected for translation, mediated by eukaryotic initiation factor 1 (eIF1) and eukaryotic initiation factor 5 (eIF5) [41]. While eIF1 promotes skipping of weak translational start sites, eIF5 increases initiation at these sites. The relative abundance of these two factors determines which start codon is used. The strongest translational context, also known as the ideal Kozak sequence context, contains a purine at the -3 position, preferably an adenosine (A), and a guanosine (G) at the +4 position, relative to the AUG start codon (Fig 3A). A weak translational context results when either of these purines at the -3 and +4 positions is substituted by a pyrimidine. The *JUN* mRNA possesses two AUG start codons, an in-frame upstream AUG (uAUG) four codons before a main AUG (mAUG), with different translational contexts (Fig 3A). The *JUN* uAUG possesses a weak translational context, with a uridine (U) at the -3 position and an adenosine (A) at the +4 position. By contrast, the *JUN* mAUG has a strong translational context, with an adenosine (A) at the -3 position and a guanosine (G) at the +4 position. It is not known which of these *JUN* AUGs is preferentially selected for translation and there currently is no evidence of JUN peptides that initiate at the uAUG.

To investigate whether *JUN* translation can initiate at either AUG or whether one is preferentially selected, we designed mRNA reporter constructs containing the *JUN* 5′ UTR and the first 51 nucleotides of the *JUN* CDS (corresponding to 17 amino acids), followed by the full Nluc CDS (Fig 3B). The WT version of this construct therefore contains both *JUN* AUG start codons and their intact translational contexts. We then mutated start codons individually or their translational context to test their roles in *JUN* translation. We transfected these mRNA reporters into HEK293T cells, together with the *HBB* 5′ UTR and Fluc CDS control, and monitored translation using luciferase assays. In general, disrupting either AUG or changing their translational context significantly represses *JUN* translation, which in turn suggests that translation can initiate at both AUGs (Fig 3C). We found that disrupting either AUG by mutation to AAG repressed *JUN* reporter translation, consistent with both AUGs contributing to *JUN* translation (Fig 3B and 3C). The more substantial decrease in *JUN* reporter translation due to the mAUG (*JUN* ΔmAUG, 95% reduction) compared to mutation of the uAUG (*JUN* ΔuAUG, 75% reduction) suggests that the mAUG start codon may be preferred in our experimental conditions.

Changing the translational context of either AUG also repressed *JUN* translation. Interestingly, making the sequence context for the uAUG stronger–either by introducing an A in the

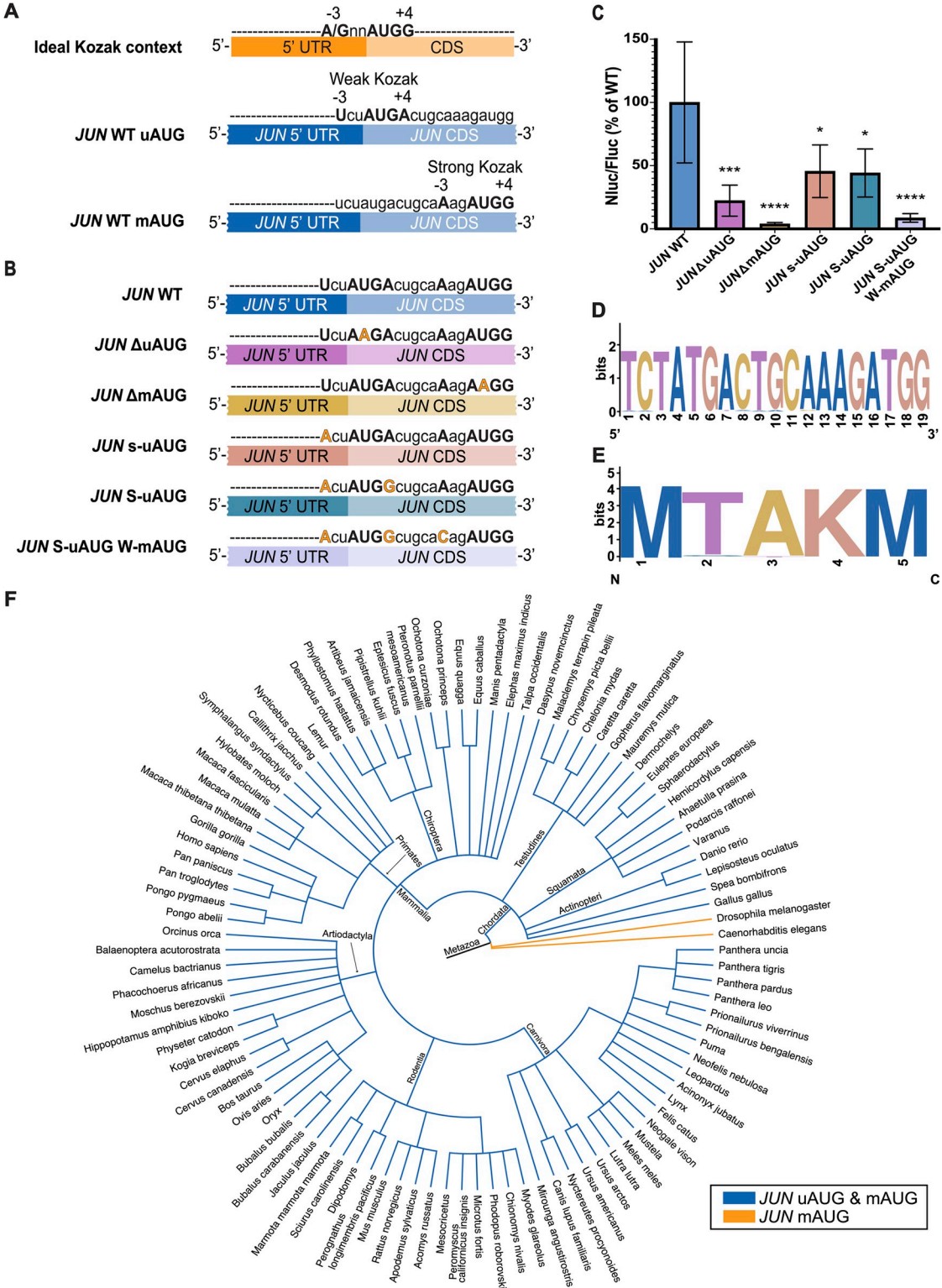

**Fig 3. Two start codons contribute to *JUN* translation.** (A) Diagram depicting the ideal Kozak context for a generic open reading frame (A/GnnAUGG). Below, diagrams depicting each of the *JUN* start codons (AUG) and their translational contexts. (B) Diagram depicting *JUN* mRNA reporter constructs, with their corresponding mutations in each of the *JUN* start codons and their translational contexts. The constructs contained the full *JUN* 5′ UTR sequence along with the first 51 nucleotides of the *JUN* CDS, upstream of the full Nluc CDS. (C) Luminescence from HEK293T cells transfected with *JUN* 5′ UTR and 51nt *JUN* CDS and Nluc

CDS reporter mRNAs (WT, ΔuAUG, ΔmAUG, s-uAUG, S-uAUG or S-uAUG W-mAUG), together with an *HBB* 5′ UTR and Fluc CDS control, assessed using a dual-luciferase assay as in Fig 1. Nluc/Fluc measurements of each mutant were normalized to the WT *JUN* 5′ UTR and 51nt *JUN* CDS and Nluc CDS measurements and reported as percentages. Technical triplicates for each biological replicate, and a total of at least three biological replicates were taken for each measurement. P values determined using a one-sample t test versus a hypothetical value of 100 are shown as follows: *p ≤ 0.05, **p ≤ 0.01, ***p ≤ 0.001, ****p ≤ 0.0001. The mean value of the replicates and standard error of the mean are shown. (D) Sequence logo depicting the conservation of the 19 nucleotide *JUN* sequence spanning both start codons and their translational context amongst 100 species. (E) Sequence logo depicting the conservation of the 5 amino acid *JUN* sequence containing both start codon methionines amongst 100 species. (F) Phylogenetic tree depicting the conservation of both *JUN* AUGs amongst 100 species. Species with both the *JUN* uAUG and the *JUN* mAUG are depicted with blue branches, while species with only one *JUN* mAUG are depicted in orange.

-3 position of the upstream AUG (Fig 3B) or by also including a G mutation in the +4 position to make it an ideal Kozak sequence–resulted in a 50% decrease in translation (Fig 3C). Moreover, using the uAUG in a strong Kozak context while weakening the translational context of the mAUG further represses *JUN* translation, to about 10% of the WT levels (Fig 3C, mutant S-uAUG W-mAUG). Taken together, these results strongly support the hypothesis that both AUGs are used for translation, and that the preference for which AUG is selected for initiation depends partly on its translational context.

### *JUN* uAUG and mAUG are conserved in vertebrates

To further investigate whether both *JUN* AUGs contribute to its translation, we examined sequence conservation of the *JUN* 5′ UTR and early CDS region that contains both AUG start codons and their translational context. We searched the 19-nucleotide region spanning the Kozak contexts of both AUGs in 100 species using the Genome Data Viewer (NLM-NCBI) and Ensembl for sequence confirmation (S1 Table). Remarkably, sequences in this region are conserved both at the nucleotide and at the amino acid level in the species examined (Fig 3D and 3E). Conservation of both *JUN* AUGs is present in all vertebrates [62], whereas only the mAUG is present in the invertebrates we investigated (Fig 3F). This conservation of both of *JUN*'s AUGs and their translational context suggests an ancient mechanism for *JUN* translation regulation and highlights the importance of both *JUN* AUGs.

### Discussion

Given that *JUN* was the first oncogenic transcription factor identified [51, 52] it is notable how little is known mechanistically about how *JUN* expression is controlled at the translational level. In this work we probed the contributions of mRNA features and initiation factors to *JUN* translation regulation in human cells. Our study reveals that *JUN* translation regulation is a complex process that is mediated by mRNA target sequences and structural elements spanning the entire *JUN* 5′ UTR (Fig 4A). Moreover, we provide evidence that initiation factors in addition to eIF3 [18] contribute to *JUN* translation regulation (Fig 4B). We also found that both the uAUG and mAUG contribute to *JUN* translation. Given that the *JUN* 5′ UTR has a length that exceeds the average 218 nt human 5′ UTR [63], a high level of secondary structure [18], and a GC rich sequence, our hypothesis is that many features within its 5′ UTR that participate in its regulation are still unknown.

Previous results found that eIF3 can directly bind structures in the 5' UTR of specific mRNA transcripts to regulate their translation, with *JUN* serving as a prototypical example [18]. Here we explored the regulatory roles of RNA structural elements within or near the eIF3-interacting stem loop (SL) region of the *JUN* 5' UTR (Fig 1). In addition to the importance of this SL for enhancing *JUN* translation in cells, we found that replacing the SL loop by a highly-stable UUCG tetraloop [59] increases *JUN* translation. It is possible that significant

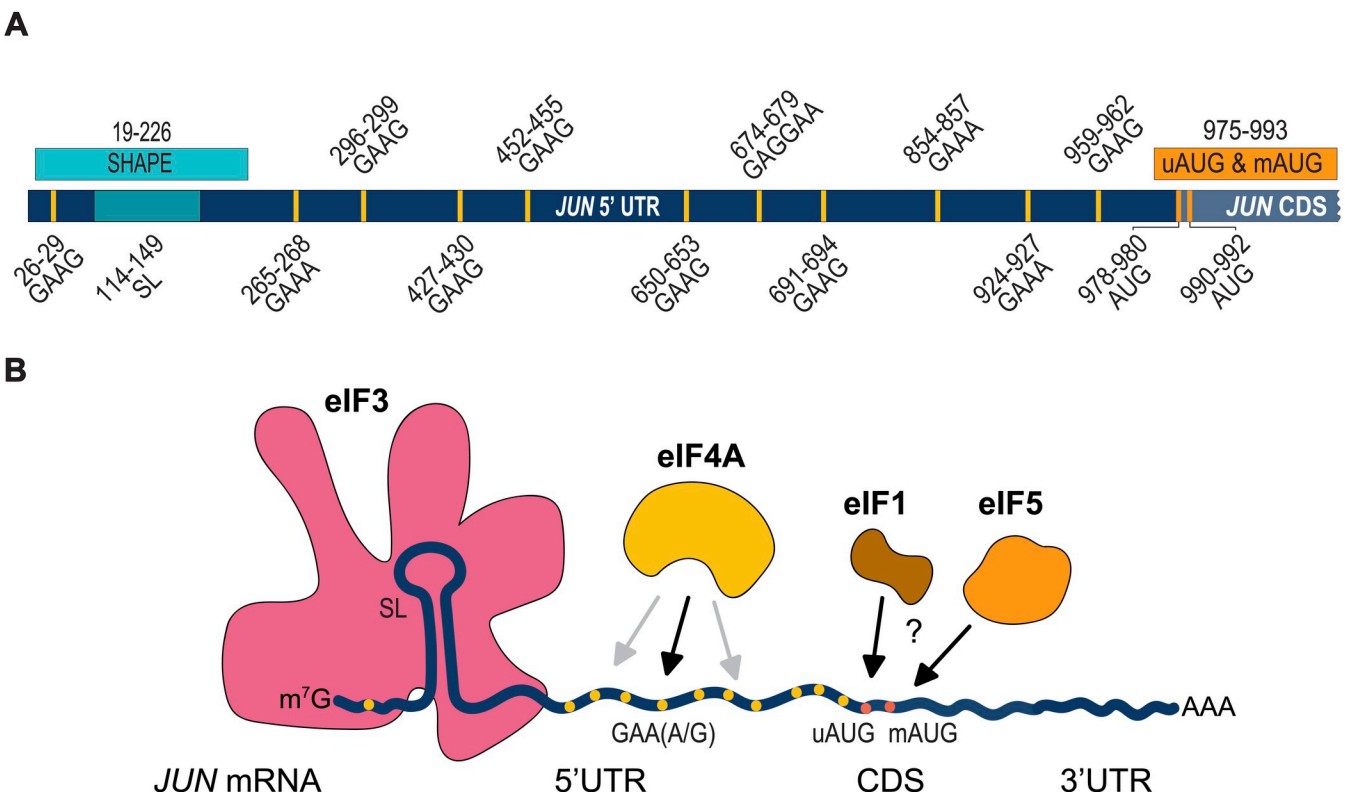

**Fig 4. *JUN* translation regulation is mediated by 5' UTR features and multiple translation initiation factors.** (A) Diagram showing all *JUN* regions investigated in this study. (B) Diagram depicting different contributors to *JUN* translation regulation. These factors include: *JUN* mRNA secondary structure (depicted as a stem loop in the 5′ UTR), *JUN* sequences (such as the GAA(A/G) polypurine sequences depicted in yellow), *JUN* AUGs (depicted in orange), and initiation factors (such as eIF3, eIF4A, and potentially eIF1 and eIF5). EIF3 (depicted in pink) interacts with structured regions of the *JUN* 5′ UTR. EIF4A (depicted in yellow) interacts with GAA(A/G) sequences (black arrow) and with other unknown sequences (gray arrows) in the *JUN* 5′ UTR. EIF1 (depicted in brown) and eIF5 (depicted in orange) may play roles in *JUN* start codon selection.

local rearrangements may be required for the canonical *JUN* SL loop sequence to bind eIF3 or that the canonical SL loop is highly dynamic, and insertion of the UUCG tetraloop locks this structure in the most favorable conformation for eIF3 binding. Additionally, there may be some sequence specificity in the SL loop, as mutation of two nucleotides in the context of the wild-type loop at positions 128 and 129 also affect *JUN* translation levels (Fig 1). Interestingly, the ability of this eIF3-interacting SL structure to promote translation is shown by the fact that it can be inserted in a modular way into the 3′ UTR of reporter mRNAs to promote translation, as shown in activated T cells [20]. We also found additional structural elements besides the eIF3-interacting SL that contribute to *JUN* translation. Most notably, these are a stem loop between nucleotides 23 and 33 of the *JUN* 5′ UTR and a bulge loop between nucleotides 42 and 47, which enhance or repress translation, respectively (Fig 1). These secondary structure elements may serve potential regulatory roles, similar to the one shown for the eIF3-interacting SL. These results are consistent with previous findings which have correlated long and highly structured 5′ UTRs with complex regulation mediated by eIF3 [58]. However, it remains to be determined whether eIF3 interacts directly with these regions. It is also possible these secondary structure elements mediate additional regulatory interactions [63]. Further evidence will be required to determine whether additional initiation factors interact with the structural elements studied in this SL-proximal region. Moreover, it would be interesting to investigate the effect of combinations of mutations in the structure and sequence elements found to influence

*JUN* translation, and how these combinations affect eIF3 binding to the *JUN* 5' UTR. Regarding the eIF3-binding region, it is important to note that *in vitro* experiments with the *JUN* 5' UTR ΔSL construct have consistently shown less robust translational repression when compared to experiments done in cells (Fig 2). Because of this, we hypothesize that the mechanism that regulates translation of this mRNA reporter construct requires elements predominantly active *in vivo*.

We also found evidence for a role for eIF4A in *JUN* translation regulation. We demonstrated that *JUN* is highly sensitive to RocA, consistent with prior transcriptome-wide experiments [33] and with *JUN* being a target of eIF4A regulation (Fig 2). Interestingly, RocA sensitivity is independent of *JUN* interactions with eIF3, since mutations in the *JUN* eIF3-interacting SL did not affect its sensitivity to RocA (Fig 2). RocA sensitivity of additional *JUN* mRNA mutations near the eIF3-binding site in the 5' UTR is an area of interest for future studies. RocA was shown to clamp eIF4A onto GAA(G/A) polypurine sequences in a subset of RocA sensitive mRNAs and these mRNAs are in fact rich in these tetramer motifs [33]. Notably, *JUN* possesses 11 of these GAA(G/A) motifs in its 5' UTR; however, mutating these sequences to CAAC did not overcome *JUN* sensitivity to RocA, suggesting that RocA may clamp eIF4A onto additional polypurine sequences in the *JUN* 5′ UTR different from the predominant motif previously identified [33]. A potential polypurine sequence present in the *JUN* 5′ UTR and to which RocA might clamp eIF4A is AGAG [34]. Within eIF3, subunit EIF3D can bind to the *JUN* mRNA 5′-7-methylguanosine cap structure, while an RNA structural element adjacent to the cap blocks recruitment of the eIF4F complex [19]. However, our results with RocA treatment suggest that at least some of the eIF4F components may contribute to *JUN* mRNA recruitment and scanning. This suggests that there may be a novel mRNA recruitment complex for *JUN*, in which eIF4A is present despite the absence of eIF4E, with EIF3D possibly acting as the cap-binding protein in this context. Our results on the involvement of eIF4A in *JUN* translation are surprising when compared to results shown in a previous study [19]. This discrepancy could be due to differences on the experimental approach. In previously reported experiments, 48S-like complexes were isolated using sucrose gradients. It is possible that these complexes represent a late stage of translation initiation and that eIF4A plays a role at an earlier stage. Future studies evaluating this will be required to dissect the step-to-step mechanism of *JUN* translation initiation.

Although *JUN* possesses a 5' UTR nearly 1 kb in length, it also has two closely-spaced potential start codons, an upstream start codon (uAUG) 4 codons away from a downstream "main" AUG (mAUG). However, which of these start codons is preferentially selected and whether they both contribute to *JUN* translation is currently unknown. Notably, experimental evidence for usage of the uAUG would be missed in published mass spectrometry experiments due to presence of a lysine at codon -1 relative to the mAUG, which would lead to removal of the leading peptide in commonly-used trypsin digests. Using reporters with the full-length *JUN* 5' UTR and both AUGs, we find that both AUGs likely contribute to *JUN* translation, albeit in a complex way (Fig 3). For example, deleting each AUG individually, repressed *JUN* translation significantly, with deletion of the mAUG causing a more severe reduction. However, the contexts of the uAUG and mAUG do not always correlate with translational output. For example, changing the weak context of the uAUG seen in WT *JUN* into a strong context decreased translation by 50% rather than increasing it. In this case, the mAUG is also likely used, as weakening the translational context of the mAUG in the strong uAUG context background further repressed translation to about 10% of WT levels. These results suggest that while both AUGs contribute to *JUN* translation, perhaps the mAUG plays a major role. These results also raise the possibility that translational efficiency of the first 4 codons including the uAUG may be lower than that of the mAUG, which would result in a lower translational output from the uAUG when it is used.

The fact that both AUGs may contribute to *JUN* translation suggests they may be part of a regulatory switch in varying cellular conditions. For example, unwinding of an RNA secondary structure downstream of an uAUG in an immune response promotes translation initiation at the mAUG of specific mRNAs in *Arabidopsis thaliana* [64]. Other cellular conditions, such as stress, starvation, or polyamine abundance could influence start codon selection [35, 39, 65, 66]. Finally, the relative abundance of eIF1 and eIF5 –which regulate the stringency of start codon selection [38, 40, 41]–could influence which *JUN* start codon is used, and thus the translational output of the *JUN* mRNA. Further experiments will be needed in order to test this hypothesis.

When exploring the evolution of *JUN*'s AUGs we found that both are conserved in vertebrates, which suggests an ancient mechanism of regulation for *JUN* by means of translational start site selection. Importantly, the translational context is also conserved for most of the examined species (Fig 3D and 3E, S1 Table), suggesting that the translational context plays a significant role in determining which start codon is selected. Our observations align with previous reports which showed that uAUGs are highly conserved in higher eukaryotes due to their roles in regulating translation initiation under regulatory circumstances [67, 68]. In addition, the evolutionary conservation suggests that more than one *JUN* polypeptide may be expressed by initiation of translation at both the uAUG and the mAUG. This type of alternative initiation has been shown previously by leaky scanning of uAUGs in a weak translational context, especially of those that are close to their downstream mAUG which allows for backward oscillation of the ribosome [69, 70]. Further studies are needed in order to test whether *JUN* leads to expression of more than one polypeptide, depending on the start codon selected. For example, this would require using a different protease for mass spectrometry besides trypsin to avoid cleavage after the lysine at position -1 relative to the mAUG, to retain N-terminal peptides originating at the uAUG.

It is notable that many different mechanisms regulate *JUN* expression at the translational level. Our study demonstrates the potential of the *JUN* mRNA as a model transcript for understanding new mechanisms of mRNA translation regulation. It opens the doors for further exploration of the regulatory roles of long and highly structured 5′ UTRs and the initiation factors that participate in translation regulation. It also points to possible new roles for *JUN* mRNA translation levels in mediating cellular response to a wide array of physiological conditions.

## Materials and methods

### Reporter plasmids

To generate the *JUN* 5′ UTR and the *HBB* 5′ UTR Nluc reporter plasmids, the *JUN* 5′ UTR (ENST00000371222.4) previously generated by amplification from human cDNA [18] and the *HBB* 5′ UTR (ENST00000335295.4) commercially generated (IDT) sequences were each inserted into the pNL1.1 *NanoLuc* luciferase reporter plasmid (Promega, GenBank Accession Number JQ437370) downstream of a T7 promoter using overlap-extension PCR with Q5 High-Fidelity DNA Polymerase (NEB) and InFusion cloning (Takara Bio). For the *JUN* AUG mutants, the first 51 nucleotides of the *JUN* CDS were inserted downstream of the full *JUN* 5′ UTR sequence and upstream of the full Nluc CDS in the pNL1.1 plasmid. For the Fluc reporter plasmid, the *HBB* 5′ UTR Nluc reporter plasmid was amplified and the *NanoLuc* luciferase sequence was replaced by a commercially generated *Firefly* luciferase sequence (IDT) [71]. Subsequent mutant versions of the *JUN* reporter plasmids were made by amplifying the plasmid using overlap-extension PCR with Q5 High-Fidelity DNA Polymerase (NEB) and primers containing the corresponding mutations, insertions, or deletions, followed by InFusion

cloning (Takara). All primers used for amplification can be found in S2 Table. All sequences were verified by Sanger sequencing. Protocol available: http://dx.doi.org/10.17504/protocols. io.kqdg3xoyzg25/v1.[PROTOCOL DOI].

### *In vitro* transcription

All RNA reporters were made by *in vitro* transcription with a standard T7 RNA polymerase protocol using DNA template gel extracted using the Zymoclean Gel DNA Recovery Kit (Zymo), 1x T7 RNA Polymerase buffer (NEB), 5 mM ATP (Thermo Fisher Scientific), 5 mM CTP (Thermo Fisher Scientific), 5 mM GTP (Thermo Fisher Scientific), 5mM UTP (Thermo Fisher Scientific), 5 µg BSA (NEB), 9 mM DTT, 25 mM $MgCl_2$, 200U T7 RNA polymerase (NEB), 50U Murine RNAse inhibitor (NEB) and incubating for 4 hours at 37°C. The DNA template used for *in vitro* transcription was generated by PCR amplification from the corresponding reporter plasmid using the Q5 High-Fidelity DNA Polymerase (NEB) with a reaction including a forward primer containing the T7 promoter sequence and a 60T reverse primer for polyadenylation. Primers used for each transcript can be found in S2 Table. After *in vitro* transcription, RNAs were treated with DNAse (Promega) following the manufacturer's protocol and precipitated with 7.5 M lithium chloride. RNAs were then capped using Vaccinia D1/ D2 (Capping enzyme) (NEB) and 2′ O-methylated using Vaccinia VP39 (2′ O Methyltransferase) (NEB) in a reaction that also included 1X capping buffer (NEB), 10 mM GTP (Thermo Fisher Scientific) and 4 mM SAM (NEB). RNAs were then purified with the RNA Clean and Concentrator-5 Kit (Zymo). In order to verify the integrity of the *in vitro* transcribed mRNAs, 6% polyacrylamide TBE-Urea denaturing gels were run using 1X TBE (Invitrogen), a ssRNA ladder (NEB) and SYBR safe stain (see representative gel in S1 Fig). Protocol available: http:// dx.doi.org/10.17504/protocols.io.ewov1qwxpgr2/v1.[PROTOCOL DOI.

### HEK293T cells and mRNA transfections

HEK293T cells were maintained in DMEM (Gibco) supplemented with 10% FBS (VWR) and 1% Pen/Strep (Gibco). Cells were grown at 37°C in 5% carbon dioxide and 100% humidity. Luciferase reporter mRNAs were transfected into these cells using the TransIT-mRNA Transfection Kit (Mirus), with the following protocol modifications. HEK293T cells were seeded into opaque 96-well plates (Thermo Fisher Scientific) about 16 hours prior to transfections. The next day, once the cells reached 80% confluency, transfections were performed by adding the following to each well: 7 µL of pre-warmed OptiMEM media (Invitrogen), 500 ng of the corresponding 5′ -capped and 3′ -polyadenylated *JUN* 5' UTR and Nluc CDS reporter mRNA, 150 ng of 5′ -capped and 3′ -polyadenylated *HBB* 5' UTR and Fluc CDS reporter mRNA, 2 µL of Boost reagent (Mirus Bio) and 2 µL of TransIT mRNA reagent (Mirus Bio). In this context, the *HBB* 5' UTR and Fluc CDS construct serves as an internal control to account for transfection efficiency and mRNA stability. Transfection reactions were incubated at room temperature for 3 minutes prior to drop-wise addition into each well. For experiments presented in Fig 2A, transfected cells were treated with increasing concentrations of RocA (+RocA) or DMSO control (+DMSO) 3 hours post-transfection, as previously reported [33]. An mRNA with the *HBB* 5′ UTR and Nluc CDS mRNA was used as a RocA-insensitive control. Transfected cells were incubated at 37°C for 8 hours, after which luciferase assays were performed using the NanoGlo Dual-Luciferase Reporter Assay System (Promega) following the manufacturer's protocol. Luminescence was then measured independently for the Nluc construct and for the Fluc construct in each sample using a Spark multimode microplate reader (TECAN). Nluc/ Fluc ratios were calculated and normalized to the corresponding control condition, set as 100%. Technical triplicates for each biological replicate, and a total of at least three biological

replicates were taken for each measurement. P values were determined using a one-sample t test versus a hypothetical value of 100. The mean value of the replicates and standard error of the mean were plotted. Protocol available: http://dx.doi.org/10.17504/protocols.io.36wgq3zyklk5/v1.[PROTOCOL DOI].

## HEK293T pSB-HygB-GADD34-K3L cells and extract preparation

HEK293T pSB-HygB-GADD34-K3L cells were maintained in DMEM media (Gibco) supplemented with 10% Tet-system approved FBS (Gibco) and 1% Pen/Strep (Gibco) [72]. Cells were grown at 37°C in 5% carbon dioxide and 100% humidity. Cells were grown for extract preparation as follows. The day after plating cells from a frozen stock into a T25 flask (Cell Star), media was exchanged and supplemented with 200 μg/mL Hygromycin B (Invitrogen). The following day, cells were transferred to a T75 flask (Corning) with media supplemented with 200 μg/mL Hygromycin B. Once cells reached 100% confluency, half of the cells were transferred to a T175 flask (Falcon) with media supplemented with 200 μg/mL Hygromycin B. Once cells reached 100% confluency, cells were passaged onto 25 150 mm plates (Corning) at a 1 to 25 ratio. The next day, cells were treated overnight with 20 μg Doxycycline (Takara Bio) per plate.

*In vitro* translation extracts were made from HEK293T pSB-HygB-GADD34-K3L cells using a previously described protocol [72]. Cells were placed on ice, scraped and collected by centrifugation at 1000 xg for 5 minutes at 4°C. Cells were washed once with ice-cold DPBS (Gibco) and collected once again by centrifugation at 1000 xg for 5 minutes at 4°C. After this, cells were homogenized with an equal volume of freshly made ice-cold hypotonic lysis buffer (10 mM HEPES-KOH pH 7.6, 10 mM KOAc, 0.5 mM Mg(OAc)$_2$, 5 mM dithiothreitol). After hypotonic-induced swelling for 45 minutes on ice, cells were homogenized using a syringe attached to a 26G needle (BD). Extract was then centrifuged at 15000 xg for 1 minute at 4°C. The resulting supernatant was aliquoted, frozen with liquid nitrogen, and stored at -80°C. Protocol available: http://dx.doi.org/10.17504/protocols.io.eq2lyjr2mlx9/v1.[PROTOCOL DOI].

## *In vitro* translation

*In vitro* translation reactions were performed using HEK293T pSB-HygB-GADD34-K3L translation-competent cell extract, as previously described [72]. Translation reactions contained 50% translation-competent cell extract, 52 mM HEPES pH 7.4 (Takara), 35 mM potassium glutamate (Sigma), 1.75 mM Mg(OAc)$_2$ (Invitrogen), 0.55 mM spermidine (Sigma), 1.5% Glycerol (Fisher Scientific), 0.7 mM putrescine (Sigma), 5 mM DTT (Thermo Scientific), 1.25 mM ATP (Thermo Fisher Scientific), 0.12 mM GTP (Thermo Fisher Scientific), 10 mM L-Arg; 6.7 mM each of L-Gln, L-Ile, L-Leu, L-Lys, L-Thr, L-Val; 3.3 mM each of L-Ala, L-Asp, L-Asn, L-Glu, Gly, L-His, L-Phe, L-Pro, L-Ser, L-Tyr; 1.7 mM each of L-Cys, L-Met; 0.8 mM L-Trp, 20 mM creatine phosphate (Roche), 60 μg/mL creatine kinase (Roche), 4.65 μg/mL myokinase (Sigma), 0.48 μg/mL nucleoside-diphosphate kinase (Sigma), 0.3 U/mL inorganic pyrophosphatase (Thermo Fisher Scientific), 100 μg/mL total calf tRNA (Sigma), 0.8 U/μL RiboLock RNase inhibitor (Thermo Scientific), and 1000 ng of the corresponding mRNA. Reactions were then incubated for 60 minutes at 32°C, and Nanoluciferase activity was monitored using the Nano-Glo Luciferase Assay Kit (Promega) using a Spark multimode microplate reader (TECAN). The average of each biological replicate was normalized to the control condition, set as 100%. Technical triplicates for each biological replicate, and a total of at least three biological replicates were taken for each measurement. P values determined using a one-sample t test versus a hypothetical value of 100 are shown as follows: *p ≤ 0.05, **p ≤ 0.01, ***p ≤ 0.001, ****p ≤ 0.0001. The mean value of the replicates and standard error of the mean

were plotted. Protocol available: http://dx.doi.org/10.17504/protocols.io.bp2l6x6yklqe/v1.
[PROTOCOL DOI].

## Conservation analysis for *JUN* AUGs

The 19-nucleotide *JUN* 5′ UTR and *JUN* CDS region that spans both AUG start codons and their translational context was searched in 100 species. Species were selected randomly, starting with *Homo sapiens* and increasing the evolutionary distance throughout the vertebrates up to the invertebrates (S1 Table). Species sequences were compiled using the Genome Data Viewer (NLM-NCBI) and Ensembl. Sequence logos for the conserved nucleotide and amino acid sequences were created using WebLogo (https://weblogo.berkeley.edu/) [73, 74]. Taxonomy analysis for the species of interest was performed using the NCBI Taxonomy Browser [62, 75]. Phylogenetic tree was generated using FigTree v1.4.4 (http://tree.bio.ed.ac.uk/software/figtree/).

## Supporting information

**S1 Fig. Representative mRNA TBE-Urea gel.** 6% TBE-Urea gel for *in vitro* transcribed mRNA for the WT or ΔSL *JUN* 5′ UTR and Nluc CDS reporter constructs. nt, nucleotide. (TIF)

**S1 Table. Conservation analysis for *JUN* uAUG and mAUG.** Compilation of species investigated for *JUN* AUGs conservation analysis, including the nucleotide and amino acid sequences of the 19-nucleotide *JUN* 5′ UTR and *JUN* CDS region that spans both AUG start codons and their translational context for each species, the corresponding Reference Sequence (RefSeq) accession numbers for each sequence, and the percent similarity of each sequence to the human *JUN* sequence. (XLSX)

**S2 Table. Primers used in this study.** Compilation of primer sequences used for cloning and *in vitro* transcription amplification of each reporter construct used in this study. (XLSX)

**S1 Data.** (XLSX)

**S1 Raw image.** (PDF)

## Acknowledgments

We thank all members of the Cate laboratory for helpful discussions; Amy S. Y. Lee and Wenfei Li for sharing plasmids encoding *JUN* 5'UTR and *JUN* 5′ UTR ΔSL; Nikolay Aleksashin for HEK293T pSB-HygB-GADD34-K3L cells; Wenfei Li, Dasmanthie De Silva, Amy S. Y. Lee, and Nicholas T. Ingolia for experimental suggestions and advice; Sona Trika and Cameron Baker for contributions to initial experiments and data exploration; Amos Nissley, Santiago Mestre-Fos, and Pooja Mukherjee for critical reading of the manuscript.

## Author Contributions

**Conceptualization:** Angélica M. González-Sánchez, Jamie H. D. Cate.

**Data curation:** Angélica M. González-Sánchez.

**Formal analysis:** Angélica M. González-Sánchez, Eimy A. Castellanos-Silva, Gabriela Díaz-Figueroa, Jamie H. D. Cate.

**Funding acquisition:** Jamie H. D. Cate.

**Investigation:** Angélica M. González-Sánchez, Eimy A. Castellanos-Silva, Gabriela Díaz-Figueroa, Jamie H. D. Cate.

**Methodology:** Angélica M. González-Sánchez, Jamie H. D. Cate.

**Project administration:** Jamie H. D. Cate.

**Resources:** Angélica M. González-Sánchez.

**Supervision:** Angélica M. González-Sánchez, Jamie H. D. Cate.

**Validation:** Angélica M. González-Sánchez, Eimy A. Castellanos-Silva, Gabriela Díaz-Figueroa, Jamie H. D. Cate.

**Visualization:** Angélica M. González-Sánchez, Eimy A. Castellanos-Silva, Gabriela Díaz-Figueroa, Jamie H. D. Cate.

**Writing – original draft:** Angélica M. González-Sánchez, Jamie H. D. Cate.

**Writing – review & editing:** Angélica M. González-Sánchez, Eimy A. Castellanos-Silva, Gabriela Díaz-Figueroa, Jamie H. D. Cate.

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
