## [Decision Letter · Decision Letter 0]

2 Jan 2024

PONE-D-23-40117JUN mRNA Translation Regulation is Mediated by Multiple 5′ UTR and Start Codon FeaturesPLOS ONE

Dear Dr. Cate,

Thank you for submitting your manuscript to PLOS ONE. After careful consideration, we feel that it has merit but does not fully meet PLOS ONE’s publication criteria as it currently stands. Therefore, we invite you to submit a revised version of the manuscript that addresses the points raised during the review process.

We look forward to receiving your revised manuscript.

Kind regards,

Madhusudan Dey, PhD

Academic Editor

PLOS ONE

Journal Requirements:

4. Thank you for stating the following in the Acknowledgments Section of your manuscript: "We thank all members of the Cate laboratory for helpful discussions; Amy S. Y. Lee and 

Wenfei Li for sharing plasmids encoding JUN 5’UTR and JUN 5’ UTR DSL; Nikolay Aleksashin for HEK293T pSB-HygB-GADD34-K3L cells; Wenfei Li, Dasmanthie De Silva, Amy S. Y. Lee, and Nicholas T. Ingolia for experimental suggestions and advice; Sona 

Trika and Cameron Baker for contributions to initial experiments and data exploration; Amos Nissley, Santiago Mestre-Fos, and Pooja Mukherjee for critical reading of the 

manuscript. This work was supported by grants R01-GM065050 and R35-GM148352."

Please remove any funding-related text from the manuscript and let us know how you would like to update your Funding Statement. Currently, your Funding Statement reads as follows: "This work was funded by grants from the National Institute of General Medical Sciences (R01-GM065050 and R35-GM148352) to J.H.D.C. https://reporter.nih.gov/

The funders played no role in the study design, data collection and analysis, decision to publish, or preparation of the manuscript."

5. Please provide a complete Data Availability Statement in the submission form, ensuring you include all necessary access information or a reason for why you are unable to make your data freely accessible. If your research concerns only data provided within your submission, please write "All data are in the manuscript and/or supporting information files" as your Data Availability Statement.

6. Please upload a copy of Supporting Information Figure/Table/etc. Supplementary Figure 1 which you refer to in your text on page 33.

Reviewers' comments:

Reviewer's Responses to Questions

**Comments to the Author**

1. Is the manuscript technically sound, and do the data support the conclusions?

Reviewer #1: Yes

Reviewer #2: Yes

Reviewer #3: Yes

2. Has the statistical analysis been performed appropriately and rigorously? 

Reviewer #1: Yes

Reviewer #2: Yes

Reviewer #3: Yes

3. Have the authors made all data underlying the findings in their manuscript fully available?

Reviewer #1: Yes

Reviewer #2: Yes

Reviewer #3: Yes

4. Is the manuscript presented in an intelligible fashion and written in standard English?

Reviewer #1: Yes

Reviewer #2: Yes

Reviewer #3: Yes

5. Review Comments to the Author

Reviewer #1: Significance:

The recruitment of eukaryotic translation initiation factors and, consequently translation initiation, are regulated by secondary and tertiary structures found in the 5’-UTR of mRNA. Human JUN mRNA contains a long 5’-UTR with high degree of secondary structure, which facilitates the recruitment of eIF3 for the translation initiation. This is a progressive extension of previous work conducted by the Cate group (Lee et al, 2015 and 2016) where they show that eIF3 binds a stem loop (SL) in 5’UTR and regulates JUN mRNA translation. In this study, authors, identified i) additional sequences adjacent to the SL in 5’UTR, ii) showed the involvement of eIF4A in the regulation of JUN mRNA translation, and iii) the mechanisms of start codon selection in JUN mRNA. However, there are specific questions and concerns that the authors should address. The detailed comments are as follows.

1. In Fig 1, authors identified several important sequences by mutagenesis, which regulate JUN mRNA translation which are interesting, especially, mutants A, F, and G. i) It will be interesting to know what happens to a double mutant of F with A or G. Do any of mutation A or G overcome the mutation F? ii) In mutant F, do eIF3 still binds to stem loop? iii) Do mutants A and G disrupt the binding of eIF3?

2. In the presence of any of the mutations mentioned in Fig 1, does overexpression of eIF4A overcome the translation repression?

3. In Fig 2A, “JUN 5’ UTR G” wrongly labelled, it is “JUN 5’ UTR E”.

4. In Fig 2A, since mutant A and G are unreactive by SHAPE, also showed significant increase in JUN mRNA translation, do any of these mutations show resistance to RocA treatment?

5. Lee et al. (2016) show that the JUN mRNA translation initiation complex lacks eIF4F components. Here, authors showed that eIF4A plays a role in JUN mRNA translation. Can you please explain this discrepancy?

6. In Fig 2C and 2D, results are not matching for DMSO treatment. Why does the ∆SL mutant not exhibit a reduction in the ‘in-vitro translation’ sample, whereas the CAAC+∆SL mutant does show a reduction?

7. Why are two start codons of JUN mRNA significant, as they are only 4 codons apart.

References:

Lee AS, Kranzusch PJ, Cate JH. eIF3 targets cell-proliferation messenger RNAs for translational activation or repression. Nature. 2015 Jun 4;522(7554):111-4.

Lee AS, Kranzusch PJ, Doudna JA, Cate JH. eIF3d is an mRNA cap-binding protein that is required for specialized translation initiation. Nature. 2016 Aug 4;536(7614):96-9.

Reviewer #2: This manuscript addresses processes that can contribute to c-Jun mRNA translation. Prior reports has documented that Jun mRNA translation is mediated by its direct engagement with eIF3. This manuscript aims to provide some more details to this translational control process, notably some features of the secondary structure of Jun mRNA, and its potential sensitivity to an anti-cancer drug RocA, which targets eIF4A. These goals are met in this study and the manuscript is clearly written and experimental conclusions are supported by rigorous experiments. There are a few concerns, none major, that should be addressed to enhance rigor and clarity.

Concerns:

1. For the translation reporter assays involving the 5'-UTR Jun, do changes in mRNA levels (e.g. stability) factor in the luciferase measurements?

2. Provide more detail justifying the RocA concentration used the study. How do the RocA treatments for Jun translation compare with bulk translation?

Reviewer #3: The manuscript by Angelica M. Golzalez-Sanchez and coauthors describes novel findings on the translational control of the JUN mRNA. It is a follow up study on previous work published by the same group in Nature in 2015. This current manuscript investigates the sequence/structure requirements of the JUN 5’-UTR for translational control. It further identifies a contribution of eIF4A for this translational control and the requirement of in-frame upstream AUGs. The regulatory elements which the authors describe for the JUN mRNA are important findings for publication, not only because JUN is an oncogenic transcription factor but also the regulation of the JUN mRNA becomes a prototype for studying other mRNAs regulated by similar factors. The studies are well done and the manuscript written very clearly.

Specific comments

1. The study is based on mRNA transfections with the assumption that the levels of the transfected mRNAs are similar among different constructs. Can the authors declare that this is true?

2. In the Roc-A experiments can the authors detect the regulation of the endogenous JUN protein levels?

3. Can the authors comment if the uAUGs may belong in specific structures in the 5’-UTR that can influence translation independently of their function as ATG codons? Although experiments can be done to show this, I find it outside of the scope of this manuscript.

4. The Introduction and Discussion are very long for a focused manuscript. However, I enjoyed reading them both.

Overall, the findings of this manuscript are interesting and should be published.

6. PLOS authors have the option to publish the peer review history of their article (what does this mean?). If published, this will include your full peer review and any attached files.

Reviewer #1: **Yes: **Jagadeesh Kumar Uppala

Reviewer #2: No

Reviewer #3: No

---

## [Author Response · Author response to Decision Letter 0]

7 Feb 2024

Please see the Response to Reviewers included with the files.

---

## [Editor Report · Decision Letter 1]

16 Feb 2024

JUN mRNA Translation Regulation is Mediated by Multiple 5′ UTR and Start Codon Features

PONE-D-23-40117R1

Dear Prof. Cate,

We’re pleased to inform you that your manuscript has been judged scientifically suitable for publication and will be formally accepted for publication once it meets all outstanding technical requirements.

Kind regards,

Madhusudan Dey, PhD

Academic Editor

PLOS ONE
---

## [Editor Report · Acceptance letter]

1 Mar 2024

PONE-D-23-40117R1 

PLOS ONE

Dear Dr. Cate, 

I'm pleased to inform you that your manuscript has been deemed suitable for publication in PLOS ONE. Congratulations! Your manuscript is now being handed over to our production team.

Kind regards, 

on behalf of

Dr. Madhusudan Dey 

Academic Editor

PLOS ONE